# A PID-Type Fuzzy Logic Controller-Based Approach for Motion Control Applications

**DOI:** 10.3390/s20185323

**Published:** 2020-09-17

**Authors:** José R. García-Martínez, Edson E. Cruz-Miguel, Roberto V. Carrillo-Serrano, Fortino Mendoza-Mondragón, Manuel Toledano-Ayala, Juvenal Rodríguez-Reséndiz

**Affiliations:** 1Facultad de Ingeniería, Universidad Autónoma de Querétaro, Querétaro 76010, Mexico; jose.gm@uaq.mx (J.R.G.-M.); ecruz30@alumnos.uaq.mx (E.E.C.-M.); roberto.carrillo@uaq.mx (R.V.C.-S.); toledano@uaq.mx (M.T.-A.); 2Laboratorio de Investigación en Control Reconfigurable, Querétaro 76120, Mexico; fmendoza@uaq.mx

**Keywords:** fuzzy control, robot, PID controller, S-curve motion profile, applied artificial intelligence

## Abstract

Motion control is widely used in industrial applications since machinery, robots, conveyor bands use smooth movements in order to reach a desired position decreasing the steady error and energy consumption. In this paper, a new Proportional-Integral-Derivative (PID) -type fuzzy logic controller (FLC) tuning strategy that is based on direct fuzzy relations is proposed in order to compute the PID constants. The motion control algorithm is composed by PID-type FLC and S-curve velocity profile, which is developed in C/C++ programming language; therefore, a license is not required to reproduce the code among embedded systems. The self-tuning controller is carried out online, it depends on error and change in error to adapt according to the system variations. The experimental results were obtained in a linear platform integrated by a direct current (DC) motor connected to an encoder to measure the position. The shaft of the motor is connected to an endless screw; a cart is placed on the screw to control its position. The rise time, overshoot, and settling time values measured in the experimentation are 0.124 s, 8.985% and 0.248 s, respectively. These results presented in part 6 demonstrate the performance of the controller, since the rise time and settling time are improved according to the state of the art. Besides, these parameters are compared with different control architectures reported in the literature. This comparison is made after applying a step input signal to the DC motor.

## 1. Introduction

Linear motor motion controllers are presented in many industrial applications, including sliding door closers, assambly, conveyor systems, electronic manufacturing, material handling, industrial test, and robotic applications [1]. Motion control is a sub-field of automation that involves controlling mechanical movements of load and it is applied directly to the actuator to manage physical variables, such as torque, acceleration, velocity and position of an axis or axes, depending of the degree of freedom (DoF) of the system [2]. Motion control is applied to avoid the stress that is produced by a fast movement and to reduce the vibrations that are caused by the high rate of change in acceleration; also, trajectories are created to reach a desired position that the actuators must achieve [3,4]. Most commercial motion controllers that are available for industrial processes are based on classic controllers, such as Proportional-Integral-Derivative (PID) controller, and they are of closed architecture [5].

The radical advances in technology have changed human life perception and intensified the problem of man-machine interaction with the use of intelligent control algorithms that are capable of translating human behavior into numerical representation applied to industrial applications. In 1965, Zadeh proposed a theory of creating and processing models that are similar to those used by a human brain, called fuzzy logic (FL) [6]. This theory was never intended for use in control systems. FL tries to emulate the imprecise human reasoning of physical processes into information that is capable of being handled by an embedded system or computer [7]. The FL has been judged over the years, because of its ability to face the complex problems with no models needed. Furthermore, FL has been implemented in problems where it was believed to be impossible to find a solution [8]. The logic applied to fuzzy systems consists of sets; a fuzzy set is a class of objects with a continuum of grades of membership, whereas a classic set is composed by true or false values [9].

FL-based controllers are more flexible and complex than PID controllers, since they cover a broader range of operating conditions and they can work with inner and external disturbances of different natures. The design of fuzzy controllers is easier than developing a model-based controller, and it is customizable, due to one being able modify the structure, rule base, and display them as a human being would do to control a system. There are a vast number of applications of FL, such as simplified control of robots [10], control of car engines [11], cruise-control for automobiles [12], prediction systems for early recognition of earthquakes [13], anti-lock braking systems [14], renewable energy systems [15], aircraft engines [16], energy allocation [17,18], demand forecasting [19], predictive maintenance [20], and material handling [21], just to mention a few.

PID, state-space controllers, artificial neural networks, and FL based controllers are the techniques applied for motion control. Huang et al. [22] proposed a nonlinear adaptive controller for a linear position tracking. The end effect of the linear induction motor is used in the design, and the impact of friction dynamics is considered to design an observer-based compensation to face the friction force. The results of this work are satisfactory, but the control algorithm was implemented in MatLab/Simulink, which is a payment program and it is necessary for a license to work with it.

Kung et al. [23] proposed a controller for a two DoF platform based on Field Programmable Gate Array (FPGA). The proposed motion control has two modules. The first module generates the motion profiles and a fuzzy controller with 49 rules. The second module performs the functions of two current vector controllers of permanent-magnet synchronous motor drives. The experimental results reported good performance in the response of the controller, even with a step signal, but the number of rules increase the performance capacity.

Boualleégue et al. [24] proposed a strategy to tune a PID-type fuzzy logic controller (FLC) using the particle swarm optimization (PSO) algorithm. The scaling factors are used as constraints of the optimization algorithm. The PID-type FLC presents a suitable response using nine rules for the inference system. The convergence of the cost function satisfies the design parameters established, it converges with only 30 iterations. The paper present two disadvantages, a Matlab/Simulink license is required to implement the algorithm and the controller tuning is offline.

Bassi et al. [25] developed a self tuning PID controller while using the PSO algorithm. The reported results show its performance as compared to a PID controller tuning by the Ziegler-Nichols method, which is an empirical approach. The PSO algorithm is used to compute the controller gains minimizing a cost function compund by the integral absolute error (IAE), integral square error (ISE), and integral of time multiplied by absolute error (ITAE). The results that are presented in this paper are simulations without experimental results displayed. Khan et al. [26] designed a 49-rules FLC using the genetic algorithm (GA) to optimize the membership functions, FL rules, and scaling gains. The GA is implemented offline to obtain the design parameters, Matlab/Simulink is used for simulation. When the controller parameters are well defined the M-files, generated by Matlab, are translated into C/C++ language to be programmed in a microcontroller to control a DC motor.

A PID-type FLC is proposed by Fereidouni et al. [27]. The configuration developed is this paper is called adaptive, because the scaling factors at output are dynamically tuned while the controller is functioning. To update the denormalization factors, a stochastic hybrid bacterial foraging particle swarm optimization algorithm is used. Furthermore, several control architectures are reported. Simulation results are only proposed using a database of 49 rules. In [28], Bejarbaneh et al. submitted an adjusting technique for a PID-type FLC based on an hybrid PSO search algorithm called PSOSCALF. The controller is applied to an inverted pendulum. The design of the PID-type FLC is based on 25 rules to compute the control signal. The simulation results demonstrate that this algorithm can be well used in non-linear plants, since the control law is obtained considering the problem as an optimization one. A possible disadvantage of these kinds of combinations, optimization algorithms, and FLC is that the control signal can be trapped in a local minimum due to this being a inner characteristic of the evolutionary algorithms or optimization algorithms [29,30]. On the other hand, the design and simulation of a self-tuning PID-type FLC for an expert heating, ventilating, and air-conditioning (HVAC) system is reported by Soyguder et al. in [31].The models of the variable flow-rate HVAC system are generated using Matlab/Simulink. In this paper, three rule bases are proposed for the variables $K_{p}$, $K_{d}$ and $K_{i}$. Each rule base is composed by 25 rules, in terms of processing this is a disadvantage, since three deffuzification stages are evaluated.

In this paper a PID-type FLC is developed and implemented in a linear platform. The control algorithm has the ability to adapt itself according to the system variations. On the other hand, the fuzzy controller can be implemented in a low-cost embedded system, since the code it is easy to modify and adapt. The control algorithm is programmed in C/C++, it means that a license is not required to reproduce it. Furthermore, a profile generator is developed to design smooth trajectories that the chart must follow. The article is divided into the following sections: Section 2 and Section 3 present a FL and S-curve motion profile background, respectively. In Section 4, the PID-type FLC is designed. Section 5 presents a design methodology to implement the control algorithm based on user experience. The results and discussion are written in Section 6. Finally, the conclusion is presented in Section 7.

## 2. Fuzzy Logic Background

Fuzzy sets were first studied by L. Zadeh. Fuzzy sets are labeled by linguistic terms. Linguistic values are transformed into numerical values to cover the overall interval of the linguistic variable according to the design [32]. Examples of linguistic variables that are used in the design of fuzzy controllers are the error, the rate of change in error, and the integral of the error [33]. The way of transforming a qualitative value into a quantitative value is necessary for real-world applications.

A fuzzy set is obtained from a crisp set that allows for elements of a universal set to belong with a certain degree to a subset. A membership function in classical sets says whether the element is part or not of the set under analysis, whereas a fuzzy membership function maps each element x∈X to a real number in the interval [0,1]. A fuzzy set is defined by Equation (Equation 1).
(1)$$\tilde{A}=\left\{\left(x,\mu_{\tilde{A}}\left(x\right)\right)|x\in X,\mu_{\tilde{A}}\left(x\right)\in \left[0,1\right]\right\}$$
where $\mu_{\tilde{A}}\left(x\right)$ is the degree of membership at *x*. Fuzzy sets interact by relations.

### 2.1. Fuzzy Relations

A relation is the correspondence that exists between two or more sets; each element of a specific set corresponds to at least one element of other sets [34]. A fuzzy relation generalizes the notation described above in one that allows for a degree of partial membership. When the correspondence of elements is matched, a new set is obtained, the Equation (Equation 2) represents a relation of subsets of fuzzy cartesian product.
(2)$$\begin{array}{cc} \tilde{R}= & \{\left(\left(x_{1},x_{2},\ldots ,x_{n}\right),\mu_{\tilde{R}}\left(x_{1},x_{2},\ldots ,x_{n}\right)\right)|\\ & \left(x_{1},x_{2},\ldots ,x_{n}\right)\in X_{1}\times X_{2}\times \ldots \times X_{n},\mu_{\tilde{R}}\left(x_{1},x_{2},\ldots ,x_{n}\right)\in \left[0,1\right]\} \end{array}$$

Given two fuzzy sets $\tilde{A}$ and $\tilde{B}$, $\tilde{A},\;\tilde{B}\subseteq X$, with the membership functions $\mu_{\tilde{A}}\left(x\right)$ and $\mu_{\tilde{B}}\left(x\right)$, the intersection $\tilde{A}\cap \tilde{B}$ is expressed in Equation (Equation 3).
(3)$$\mu_{\tilde{A}\cap \tilde{B}}\left(x\right)=\min\left[\mu_{\tilde{A}}\left(x\right),\mu_{\tilde{B}}\left(x\right)\right]\;\forall x\in X$$

For fuzzy union, the fuzzy sets $\tilde{A}$ and $\tilde{B}$, $\tilde{A},\;\tilde{B}⊇X$, with the membership functions $\mu_{\tilde{A}}\left(x\right)$ and $\mu_{\tilde{B}}\left(x\right)$, the union $\tilde{A}\cup \tilde{B}$ is expressed in Equation (Equation 4).
(4)$$\mu_{\tilde{A}\cup \tilde{B}}\left(x\right)=\max\left[\mu_{\tilde{A}}\left(x\right),\mu_{\tilde{B}}\left(x\right)\right]\;\forall x\in X$$

A generalization of *n*-ary fuzzy intersection, $\tilde{A}\cap \tilde{B}$, where $\tilde{R}$ and $\tilde{S}$ are fuzzy relations, $\tilde{R},\;\tilde{S}\;\subseteq \;X_{1}\times X_{2}\times \ldots \times X_{n}$ is given by Equation (Equation 5). Similarly, the union of two fuzzy sets, $\tilde{R}\cup \tilde{S}$, for *n*-ary fuzzy relations is given by the relation of $\tilde{R}$ and $\tilde{S}$, $\tilde{R},\;\tilde{S}\subseteq X_{1}\times X_{2}\times \ldots \times X_{n}$. Equation (Equation 6) represents the union of several fuzzy sets.
(5)$$\begin{array}{cc} \mu_{\tilde{R}\cap \tilde{S}}\left(x_{1},x_{2},\ldots ,x_{n}\right)= & \min[\mu_{\tilde{R}}\left(x_{1},x_{2},\ldots ,x_{n}\right),\mu_{\tilde{S}}\left(x_{1},x_{2},\ldots ,x_{n}\right)]\\ & \forall \left(x_{1},x_{2},\ldots ,x_{n}\right)\in X_{1}\times X_{2}\times \ldots \times X_{n} \end{array}$$
(6)$$\begin{array}{cc} \mu_{\tilde{R}\cup \tilde{S}}\left(x_{1},x_{2},\ldots ,x_{n}\right)= & \max[\mu_{\tilde{R}}\left(x_{1},x_{2},\ldots ,x_{n}\right),\mu_{\tilde{S}}\left(x_{1},x_{2},\ldots ,x_{n}\right)]\\ & \forall \left(x_{1},x_{2},\ldots ,x_{n}\right)\in X_{1}\times X_{2}\times \ldots \times X_{n} \end{array}$$

The fuzzy composition is known as Max–Min composition for discrete systems. For two fuzzy relations, $\tilde{R}\subseteq X_{1}\times X_{2}$ and $\tilde{S}\subseteq X_{1}\times X_{2}$ with membership functions $\mu_{\tilde{R}}\left(x_{1},x_{2}\right)$ and $\mu_{\tilde{S}}\left(x_{2},x_{3}\right)$, the membership function $\mu_{\tilde{R}\circ \tilde{S}}\left(x_{1},x_{3}\right)$ of the composition $\tilde{R}\circ \tilde{S}$ is defined in Equation (Equation 7).
(7)$$\mu_{\tilde{R}\circ \tilde{S}}\left(x_{1},x_{3}\right)=\max_{x_{2}\in X_{2}}\min\left[\mu_{\tilde{R}}\left(x_{1},x_{2}\right),\mu_{\tilde{S}}\left(x_{2},x_{3}\right)\right]\;\;\forall \left(x_{1},x_{3}\right)\in X_{1}\times X_{3}$$

### 2.2. Membership Functions

The trapezoidal, triangular and gaussian functions are the most used membership functions used FL applications. The triangular membership function is compound by a positive and negative slope connected when the membership degree is equal to one. This function is very suitable to define situations, in which there is a central optimal value, which is lost as it moves away from it.
(8)$$\mu \left(x\right)=\left\{\begin{array}{cc} \frac{x-a}{m-a} & x\in [a,m]\\ \frac{x-m}{m-b} & x\in (m,b]\\ 0 & \;\mathrm{other}\;\mathrm{values} \end{array}\right.$$

From *a* to *b*, the triangular function is drawn. When μ(x)=m the unit is reached. In symmetric functions *m* represent the middle point. The trapezoidal membership function has an interval where μ(x) is constant and its membership degree is one. It is considered as a generalization of the triangular membership function and it is applied in systems where there is a range of optimal values, around which the conditions are not suitable. Equation (Equation 9) describes the trapezoidal function.
(9)$$\mu \left(x\right)=\left\{\begin{array}{cc} \frac{x-a}{b-a} & x\in [a,b]\\ 1 & x\in (b,c]\\ \frac{x-d}{d-c} & x\in (c,d]\\ 0 & \;\mathrm{other}\;\mathrm{values} \end{array}\right.$$

In this case, *a* and *d* are the values on which the function is depicted. On the other hand, *b* and *c* are the limits where μ(x) is the unit. For *x* values out the range [a,d], the membership function is equal to zero. The Gaussian membership function transforms the crisp values into a normal distribution and its middle point determines an ideal set, which is assigned a one. The degree of membership of the rest of the input values decreases as they move away from the midpoint, both in the positive and negative directions. This function is useful whether the membership degree is close to a specific value. The Gaussian function has similar behaviour than the triangular function, the difference is that the first one is used when it is required that the membership degrees have slow variations [35]. Equation (Equation 10) is used to compute the membership degrees of the Gaussian function.
(10)$$\mu \left(x\right)=e^{-k{\left(x-m\right)}^{2}}$$
where *m* is the middle point, *k* must be greater than 0 (k>0), and negative values are not allowed.

### 2.3. General Model of a FLC

Figure 1 depicts the general model of the FLC. A FLC is composed by four blocks (fuzzifier, rule base, inference engine, and defuzzifier), which they interact each other using fuzzy sets and fuzzy relations to have a control signal.

The fuzzifier transforms a crisp value into a fuzzy value. The information can be presented in a discrete form while using the fuzzy sets. The discretization process performs a scale mapping to transform values measured in the variables to values of the discrete universe, either uniformly or non-uniformly, or a combination of both.When the system states are available for measurement and control, the rule base are written in terms of the state variables instead of error and its derivatives [36]. In general terms, this stage contains all of the information of the application to be controlled, as well as the goals of the controller.The inference engine combines the fuzzy if-then rules for mapping the set $\tilde{A}$ from the controller input space *A* to a fuzzy set $\tilde{B}$ in the controller output space *B* using the production rules and the knowledge base of membership functions. All of the fuzzy rules are combined in a single fuzzy relation using Equation (Equation 7).The defuzzification module changes from one domain to another the sets. It means that the fuzzy numbers are transformed into crisp values according to the method to use. In the literature, there exists several ways to map from one domain to another. Once the crisp number is obtained, it is sent to the electronic interface to send it to the actuator.

## 3. S-Curve Profile

This section presents the background of the S-curve velocity profile. In Figure 2, the position, S-curve velocity profile, trapezoidal acceleration, and square jerk are displayed. As is well known, the S-curve is a seven-segment profile, where the total time displacement can be symmetric or asymmetric distributed. The acceleration and deceleration phases have the same length of time, $T_{acc}=T_{dec}$ and they are represented by three stages. The design of a trajectory is based on the desired position $\Theta_{d}$ and the time displacement *T*. The maximum velocity reached by the profile is computed in Equation (Equation 11).
(11)$$\Omega_{max}=\frac{\Theta_{d}}{(1-\Psi )T}$$

The constant factor Ψ is proposed in order to vary the duration of acceleration phase $T_{acc}=\Psi T$, which is limited by $0\leq \Psi \leq \frac{1}{2}$. The maximum magnitude of the acceleration is obtained from Equation (Equation 12). In Figure 2, one can notice that the jerk shape is modeled by a square signal placed upon the acceleration variations.
(12)$$A_{max}=\frac{\Theta_{d}}{\Psi \left(1-\Psi \right)\left(1-\eta \right)T^{2}}$$
where η is a constant factor used to calculate the length of the jerk pulse, $T_{jerk}=\eta T_{acc}$. The duration of $T_{jerk}$ must be less than $T_{acc}$ to ensure continuity. The constant jerk value is calculated in Equation (Equation 13).
(13)$$J_{max}=\frac{\Theta_{d}}{T_{jerk}\Psi \left(1-\Psi \right)\left(1-\eta \right)T}$$

Once the jerk constant value is computed, the acceleration can be calculated for any instant of time while using the Euler integral method of the Equation (Equation 14).
(14)$$A\left(t\right)=A\left(t-1\right)+\int_{t}^{T}J_{max}\left(t\right)dt$$

The S-curve velocity profile is obtained from Equation (Equation 15).
(15)$$\Omega \left(t\right)=\Omega \left(t-1\right)+\int_{t}^{T}A\left(t\right)dt$$

The desired position is reached integrating the velocity. The Equation (Equation 16) is used to compute the position.
(16)$$\Theta \left(t\right)=\Theta \left(t-1\right)+\int_{t}^{T}\Omega \left(t\right)dt.$$

## 4. PID-Type FLC Design

The PID controller is commonly used in several industrial processes and it is presented in Equation (Equation 17). A big disadvantage of this algorithms consist in the computing of the controller gains [37,38]. Figure 3 depicts the methodology proposed to compute the PID controller gains while using a fuzzy logic system. It basically consists of measuring the error, which is presented as a crisp value, and fuzzifier in order to inferred the $K_{p}$ value and then multiplied by the error signal. The same procedure is applied to the $K_{d}$. The change of rate in error is the input to the fuzzifier selected to compute $K_{d}$. Variables $K_{p}$ and $K_{d}$ offer a fast response to the plant, the term $K_{i}$ is constant, since it is only required to reduce the error to zero when the steady state is reached.
(17)$$u\left(t\right)=k_{p}e\left(t\right)+k_{d}\frac{de(t)}{dt}+k_{i}\int_{0}^{t}e\left(\tau \right)d\tau$$

The labels of the linguistic-values are presented in Table 1 with their respective ranges. Seven-linguistic-values are selected to compute the $K_{p}$ and $K_{d}$ gains.

Figure 4 presents the linguistic variable of the error and change of rate in error; they are composed by seven-linguistic-values.

Figure 4a presents an interval that goes to [−1, 1] m for error, this range is sufficient, since a trajectory is applied and the variation tends to small values upon the set point, also a step input can be applied among this interval. The linguistic variable, that represent the change of rate of error is depicted in Figure 4b and its domain goes to [−10, 10] m/s, this range of values are selected since the change of rate in error can present a high magnitude if a step input is applied. Both of the variables are composed by triangular and gaussian membership functions.

Figure 5a shows the range of values that the $K_{p}$ variable can take. Notice that these parameters cannot present negative values. $K_{p}$ can be computed in a range of [0.5, 10], according to the measured error. On the other hand, $K_{d}$ has values from a small range [0.0, 0.5]. This consideration is proposed, since the $K_{d}$ gain in a PID controller can increase the error variation and make unstable the system.

Table 2 presents the location of the singletons used to calculate the controller gains and their respective linguistic values.

Table 3 displays the rule base used for the computation of the controller gains. The total number of rules used for computing $K_{p}$ and $K_{d}$ gains is seven. Besides, one can see in Table 3 the relationship between the measured input signal, with its respective linguistic variables, and the output relationship. The inference process corresponds to a one-to-one relationship. Furthermore, it can be seen that four-input linguistic variables NM, NS, PS, and PM point to a single output variable S. These fuzzy relationships are used in order to ensure that the earnings values work on a range of adequate values.

The singleton is used as a membership function in the defuzzification stage to reduce the computational cost at the moment to compute $K_{p}$ and $K_{d}$. This is important, since, in the conventional fuzzy controller, just one defuzzification stage is required, whereas, in this proposal, two defuzzification stages are implemented to simultaneously compute both controller gains.

Once the values of the relations are founded, Equation (Equation 18) is used to transform the fuzzy set into the crisp value of the control gains. One can see in Figure 3 that $K_{p}$ and $K_{d}$ values are independently obtained.
(18)$$K_{p,d}\left[n\right]=\frac{\sum_{i=1}^{n}\mu_{c}\left(z_{i}\right)\cdot z_{i}}{\sum_{i=1}^{n}\mu_{c}\left(z_{i}\right)}$$
where $K_{p,d}\left[n\right]$ represents the controller gains, $\mu_{c}\left(z_{i}\right)$ is the membership degree of the singleton, and $z_{i}$ is the position of the singleton in the output variables $K_{p}$ and $K_{d}$. All of the possible values computed from $K_{p,d}\left[n\right]$ are depicted in Figure 6. The use of gaussian and triangular functions in the fuzzification stage are considered for approximating the response into a linear representation of all possible values that the gains must take. Additionally, it can be observed that the gains never present negative values, even if error is negative.

## 5. Design Methodology

The Figure 7 shows the steps to implement the control algorithm based on user experience. The first step consist of recognizing the parameters of the motor as the maximum voltage, current, torque, and velocity. In some cases, these parameters are not available; when this happens, experimentation is required to detect the values to start the system design [5]. When the variables needed to system design are known, the next step is to select the embedded system to interface the computer, where is the control algorithm be implemented, and the electronic platform.

The next three steps correspond to a loop, which is used to adjust the operation range for the fuzzification method and select the appropriate membership functions, generally the trapezoidal and triangular membership functions are used in microcontrollers due to their ease of processing. This is important, since the controller inputs must be well defined to be capable to perform the signals that will be mapped to fuzzy values.

The control algorithm is implemented when the ranges of operation are proposed. The algorithm computes the error and the derivative of error to be used as input to the control system. These signals are related to generate the system response by the defuzzifier and send it to the embedded system. The embedded system process the control signal and generates a Pulse-Width-Modulation (PWM) signal to move the shaft of the motor. The voltage applied to the motor does not have to exceed the maximum value permitted and proposed in first step. On the other hand, if there is not a control signal generated, then it is possible that the ranges of operation cannot detect the input signals and it is necessary to adjust the ranges until the control signal is generated. Finally, if the control signal presents a response that meets the proposed requirements, those working ranges are maintained and the control algorithm will find the gain values of the controller.

## 6. Results and Discussion

The algorithms are implemented in a HP-Omen computer (Intel Core I7 8th Generation, Santa Clara, CA, USA) and programmed in C/C++. An FPGA is used to measure the position of the encoder and send it to the computer using the RS232 communication protocol. Additionally, a PWM module was implemented in the FPGA to apply the control signal to the servo amplifier. The sample time is 0.005 s.

The linear motion system a BLM-N23-50-1000-B brushless motor was mechanically coupled whose torque constant is $K_{m}=0.08$ Nm/A with a current and continuous torque of 4.9 A and 3.9 Nm, respectively. This system has a maximum angular speed of 5000 Revolutions Per Minute (RPM) and an encoder with a resolution of 1024 Pulse Per Revolution (PPR). Figure 8 displays the linear platform used for test the control performance.

A step input signal of magnitude 0.064 m, this value is proposed, since the maximum voltage of the motor does not be exceed when the algorithm is running, is applied to the system to prove the controller performance when it is applied to the plant. Furthermore, an integral constant $K_{i}=3$ is used for the step input signal. Regularly, this type of test signals are the most used to test the response of the controllers to sudden changes in the reference signal, giving a general overview of what would happen whether a load is added to the plant. If the controller, which is faced with an aggressive test input, is not well tuned, the control response would fall into a zone of instability, which could cause permanent damage to the mechanical and electrical structure. In Figure 9a, it can be observed that the controller presents a fast response, since the system reaches the steady state in 0.248 s and the overshoot is 8.985%. Figure 9b shows the error signal and it tends to zero.

The control signal, Figure 10, is the voltage applied to the system. The maximum voltage value obtained is beneath 24 V. When a controller is applied in a real application, the most important parameter to limit is the control signal, since, if the system demands a greater quantity of the variable that is being controlled than allowed, the control algorithm must be capable of protecting itself to avoid generating values not allowed by the plant.

Figure 11a shows the behaviour of $K_{p}$ gains for a step input signal. The maximum value obtained is $K_{p}=8.43$ and is decreasing until it reaches a constant value $K_{p}=5.06$. On the other hand, $K_{d}$ reaches a maximum magnitude of 0.225, after that, it tends to converge to zero, since there is not variations in the shaft of the DC motor.

Table 4 shows the performance of different control architectures comparing the rise time, overshoot, and settling time. The response of the PID-type FLC presented in this paper has a fast response, since the first time the position signal crosses the reference for the first time is in 0.124 s and the time that it takes for the system to stabilize is 0.248 s.

### Motion Control Implementation

A S-curve velocity profile is implemented to reach a desired position instead of using a step signal. Motion control systems are compounded by a motion profile to avoid stress and reduce vibrations in the mechanical structure [41]. The cart has to move 0.53 m in 2 s, these parameters were chosen to test the controller performance; one can introduce another set-point. For smooth inputs the integral constant used is $K_{i}=8$. In Algorithm 1, the psudocode is presented to illustrate how the algorithm in C/C++ language is programmed.



Figure 12a,b present the position and error signals, respectively. For Figure 12a, the position follows a reference to compare the desired trajectory against the system position. The amplitude error oscillates between $1.8\times 10^{-5}$ and $-1.78\times 10^{-5}$ m. Note in Figure 12b that the error is zero at 2.65 s.

In Figure 13b, the control signal shows a maximum voltage peak of 16 V to achieve the desired position. The control signal is computed by the PID-like FLC relation of the controller gains. On the other hand, according to Figure 13a, one can observe that the velocity follows the reference properly, the maximum velocity computed by the trajectory algorithm is 0.94 m/s.

The controller gains tend to vary according to the error measure and rate of change of error. The variation when the controller follows as the trajectory oscillates from 3 to 6.5 in magnitude for $K_{p}$ and from 0 to 0.052 for $K_{d}$. Notice in Figure 14 that the controller gains do not have units.

In real conditions, the servo systems are in contact with loads that must move smoothly in order to prevent what they are transporting from falling; for this reason, a cylindrical load of 1.9 kg is added on the carriage and, thus, prove that the control algorithm complies with the proposed specifications. In addition, the responses obtained when the load is transported are compared, again the S-curve movement profile and the parameters that were used in the previous test are implemented, in relation to when there is no load on the cart. As one can see, the trajectory presented a similar behaviour in error whether it is compared with the signal from Figure 13b; this means that the controller is robust and it can adapt itself to variations of the system. The position follows properly the trajectory proposed, as one can see in Figure 15.

The velocity follows the trajectory properly, as one can see in Figure 16a, in stages when the acceleration and deceleration are computed, but when it must be constant, the velocity present a transient response that is induced by the load applied to the shaft of the motor. The control signal increased up to 23 V and oscillated under this value, as presented in Figure 16b.

One can see in Figure 17 that the controller gains contain aggressive oscillations after the load is applied as compared with the behaviour when no load is added, as in Figure 14, it can be justified according to the voltage response from Figure 16b.

Table 5 displays a brief comparison of motion controllers presented in the literature. As can be seen, some authors choose to use Matlab/Simulink and LabVIEW NI for the design of their algorithms, the problem with using this type of tools is that you must have a license to be able to use the software in a complete way, increasing the cost, in monetary terms, of the control system and, therefore, the algorithm is restricted to the use of embedded systems compatible with these programs. The idea of using a free programming language, which does not need licenses, for the design of motion control systems makes the system affordable and easy to apply in other programming languages and even different embedded systems. In this paper, three C/C++ functions were created in order to compute the motion profile, the membership functions, and PID-type FLC, respectively. Algorithm 1 depicts the real structure of the motion control algorithm.

## 7. Conclusions

This paper presented a new low computational cost control algorithm to tune a PID controller to motion control applications in machinery. The auto-tune algorithm is based on fuzzy logic and it is straightforward and easy to implement and translate to other computing languages. A linear platform compounded by a DC motor connected to an endless screw and an incremental encoder is used to carry out the corresponding tests. The design of the control algorithm is based on user experience, so one can propose different membership functions for the fuzzyfication stage following the steps that are presented in Figure 7. The presented results show that the PID-type FLC can modify its gains to reduce the steady error under the 0.09%, even when a load is applied to the system. When a load of 1.9 kg is added on the cart of the linear platform, the controller presented a similar response in the velocity and position. The control signal increases its magnitude from 16 to 23 V, but it is still under the maximum value permitted. On the other hand, the controller gains considerably vary its magnitude, but, when the system reaches the desired position, both gains tend to the same value obtained when the system did not present load, $K_{p}=5.2$ and $K_{d}=0$.

## Figures and Tables

**Figure 1 sensors-20-05323-f001:**
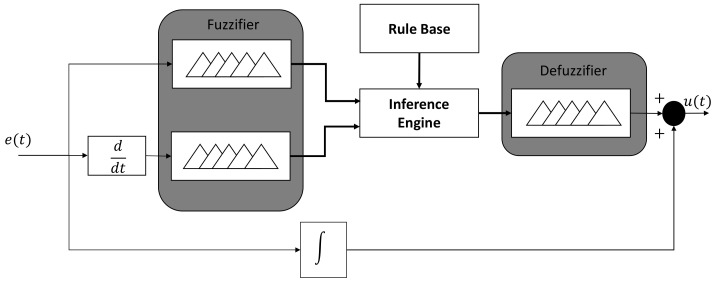
General model of the fuzzy logic controller (FLC).

**Figure 2 sensors-20-05323-f002:**
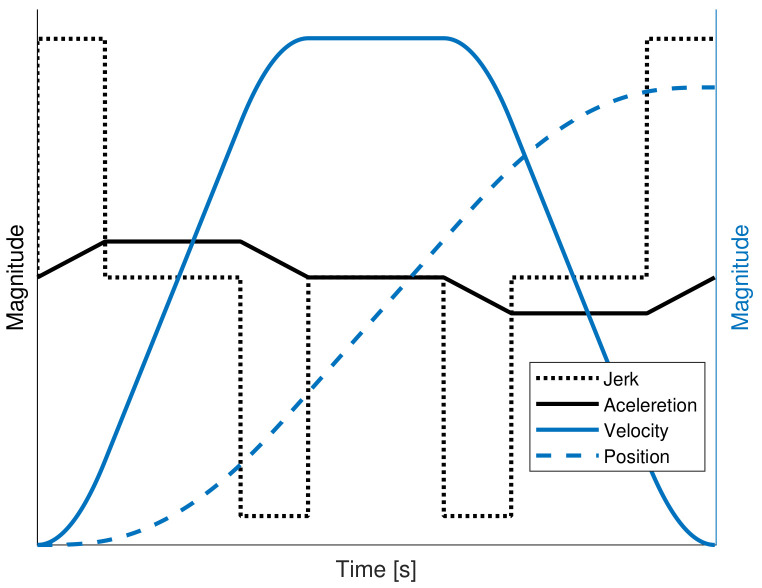
S-curve profile.

**Figure 3 sensors-20-05323-f003:**
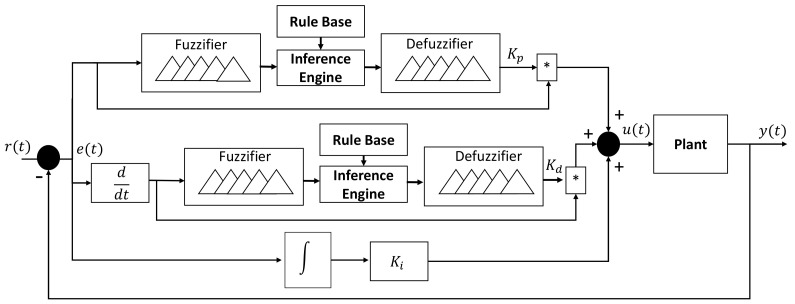
Proportional-Integral-Derivative (PID)-type FLC architecture proposed.

**Figure 4 sensors-20-05323-f004:**
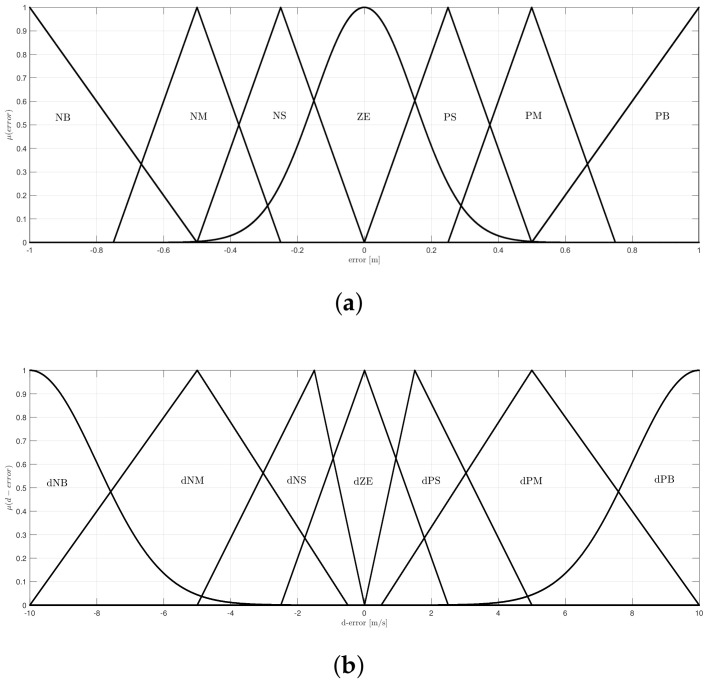
Linguistic variables (**a**) error and (**b**) derived of the error.

**Figure 5 sensors-20-05323-f005:**
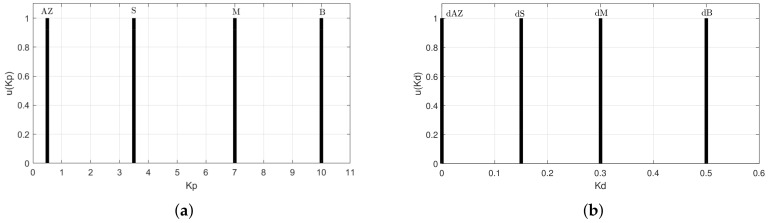
Controller gains (**a**) $K_{p}$ and (**b**) $K_{d}$.

**Figure 6 sensors-20-05323-f006:**
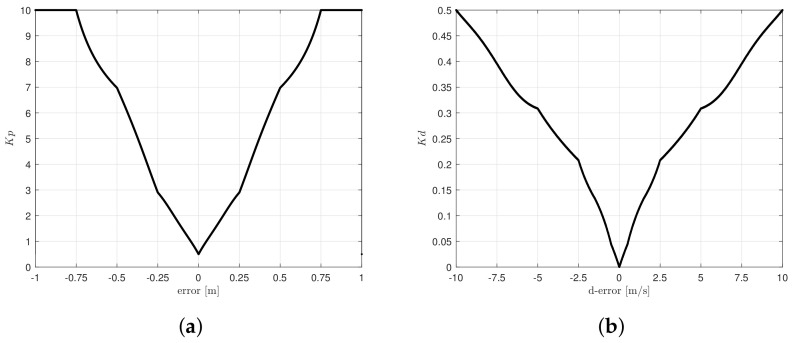
Ouput fuzzy surfece for (**a**) $K_{p}$ and (**b**) $K_{d}$.

**Figure 7 sensors-20-05323-f007:**
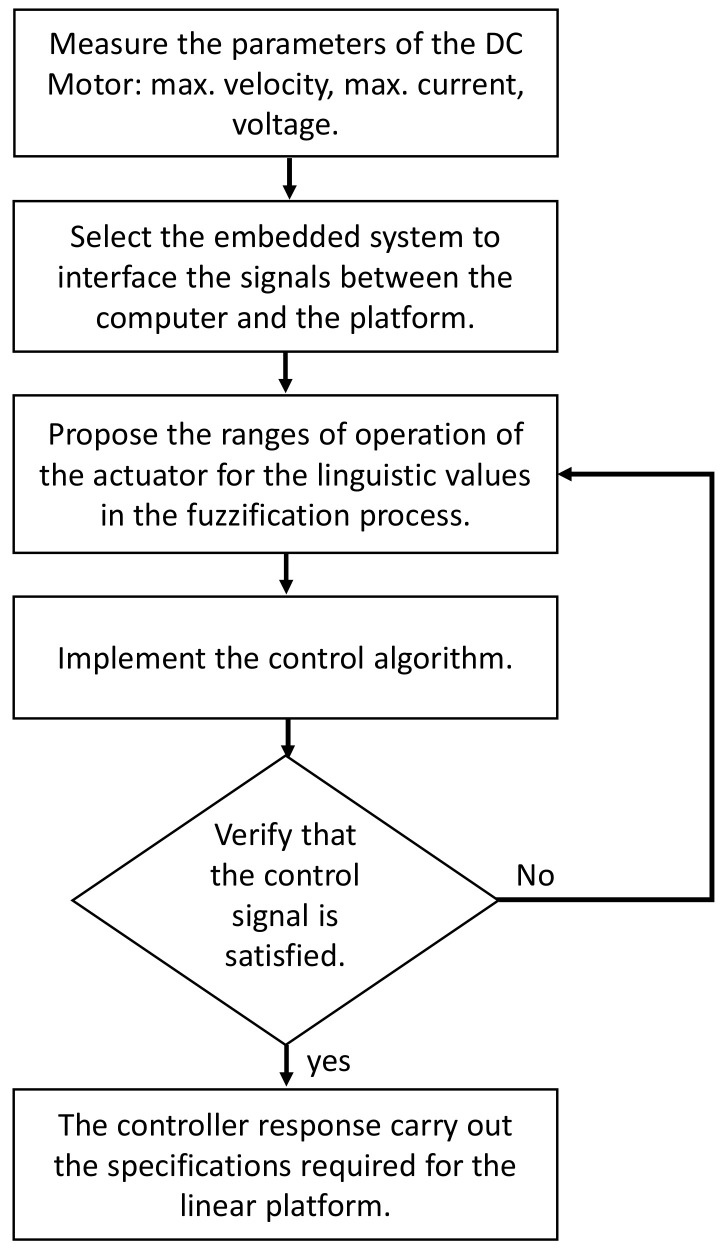
Steps to design the controller.

**Figure 8 sensors-20-05323-f008:**
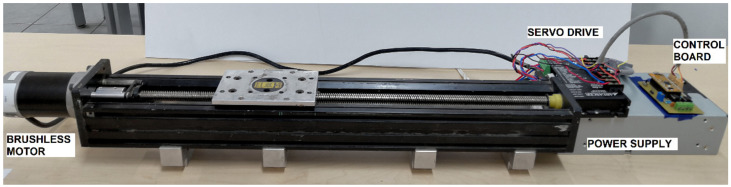
Linear platform to test the control algorithm.

**Figure 9 sensors-20-05323-f009:**
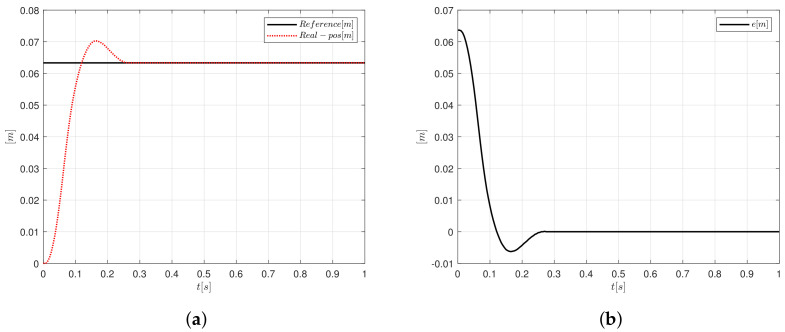
PID-like FLC response using a step signal (**a**) position and (**b**) error.

**Figure 10 sensors-20-05323-f010:**
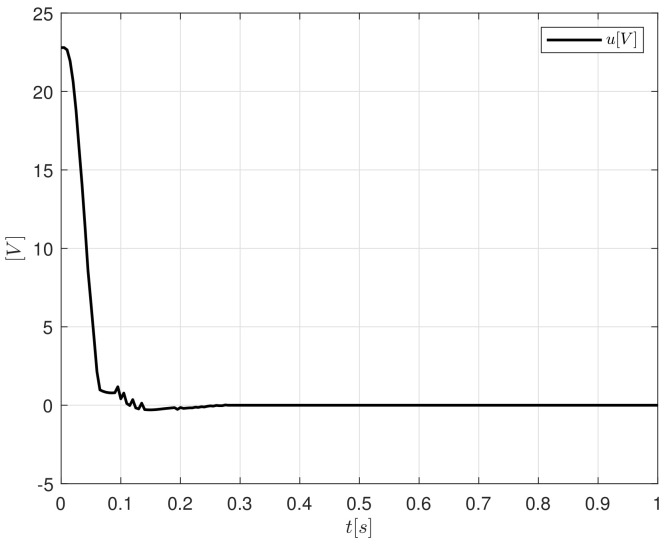
Control signal generated by PID-like FCL implementation.

**Figure 11 sensors-20-05323-f011:**
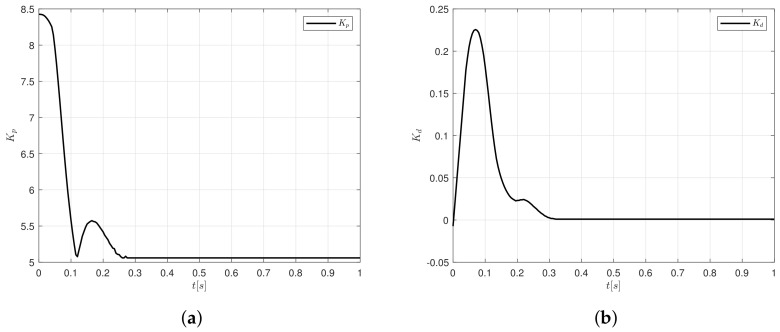
Controller gains response for the step signal (**a**) $K_{p}$ and (**b**) $K_{d}$.

**Figure 12 sensors-20-05323-f012:**
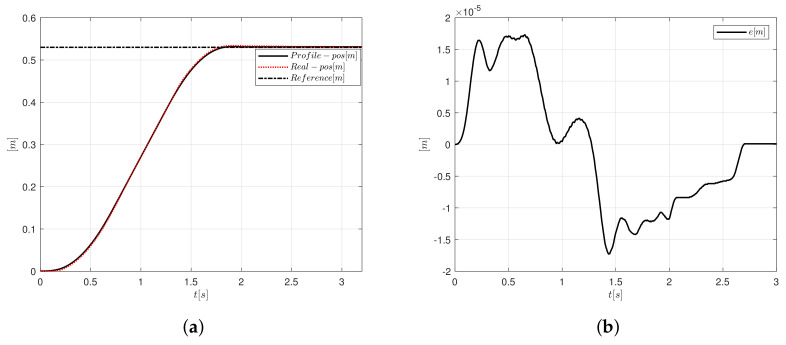
Motion controller based on PID-like FLC, and S-curve velocity profile implementation, (**a**) position and (**b**) error.

**Figure 13 sensors-20-05323-f013:**
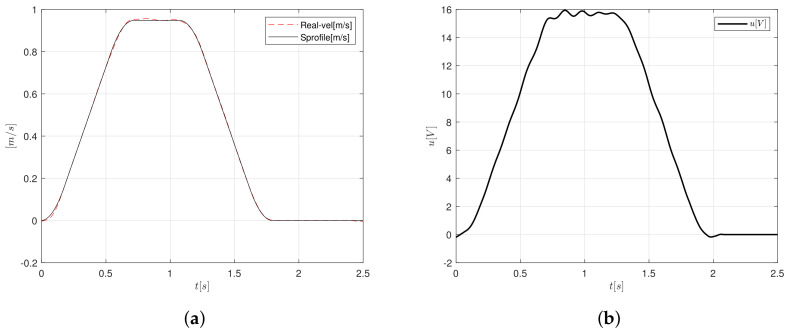
S-curve velocity profile implementation (**a**) velocity and (**b**) control signal.

**Figure 14 sensors-20-05323-f014:**
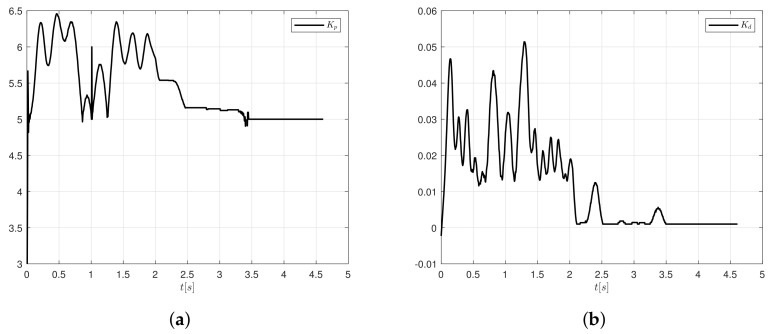
Real time response of the controller gains (**a**) Kp and (**b**) $K_{d}$.

**Figure 15 sensors-20-05323-f015:**
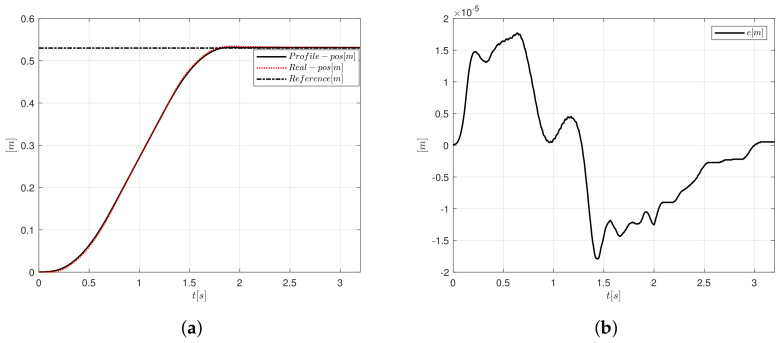
Motion controller based on PID-like FLC and S-curve velocity profile implementation adding a cylindrical load of 1.9 kg, (**a**) position and (**b**) error.

**Figure 16 sensors-20-05323-f016:**
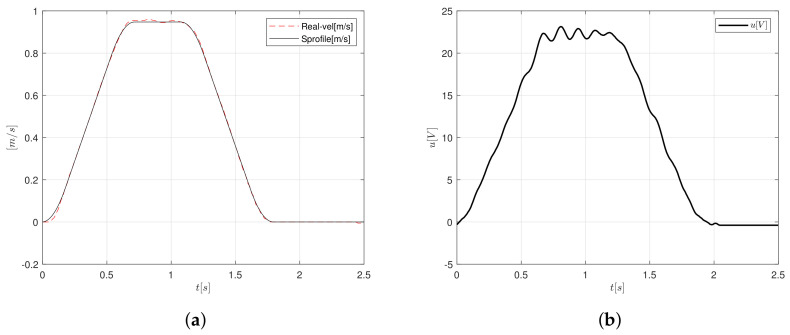
S-curve velocity profile implementation adding a cylindrical load of 1.9 kg (**a**) velocity and (**b**) control signal.

**Figure 17 sensors-20-05323-f017:**
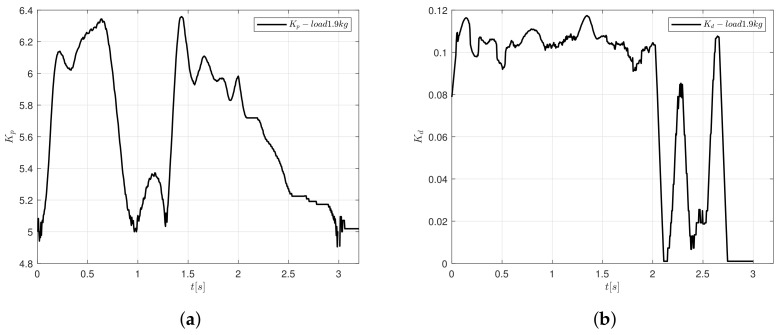
Real-time response of the controller gains adding the cylindrical load of 1.9 kg (**a**) Kp and (**b**) $K_{d}$.

**Table 1 sensors-20-05323-t001:** Linguistic values for range of error and the derived of error.

Label	Linguistic Value	Range for Error (m)	Range for d-Error (m/s)
NB	Big Negative	[−1,−0.5]	[−10,−3.50]
NM	Medium Negative	[−0.75,−0.25]	[−10,−0.50]
NS	Small Negative	[−0.50,0]	[−5,0]
ZE	Zero	[−0.50,0.50]	[−2.50,2.50]
PS	Small Positive	[0,0.50]	[0,5]
PM	Medium Positive	[0.25,0.75]	[0.5,10]
PB	Big Positive	[0.50,1]	[3.50,10]

**Table 2 sensors-20-05323-t002:** Linguistic values for $K_{p}$ and $K_{d}$.

Label	Linguistic Value	$K_{p}$	$K_{d}$
AZ	Almost zero	0.5	0.0
S	Small	3.5	0.15
M	Medium	7	0.3
B	Big	10	0.5

**Table 3 sensors-20-05323-t003:** Controller gains rules.

$K_{p}$	$K_{d}$
if e(t) is NB, then $K_{p}$ is B	if de(t) is dNB, then $K_{d}$ is dB
if e(t) is NM, then $K_{p}$ is S	if de(t) is dNM, then $K_{d}$ is dM
if e(t) is NS, then $K_{p}$ is S	if de(t) is dNS, then $K_{d}$ is dS
if e(t) is ZE, then $K_{p}$ is AZ	if de(t) is dZE, then $K_{d}$ is dAZ
if e(t) is PS, then $K_{p}$ is S	if de(t) is dPS, then $K_{d}$ is dS
if e(t) is PM, then $K_{p}$ is S	if de(t) is dPM, then $K_{d}$ is dM
if e(t) is PB, then $K_{p}$ is B	if de(t) is dNB, then $K_{d}$ is dB

**Table 4 sensors-20-05323-t004:** Performance comparison among controllers.

Work	Rise Time (s)	Overshoot (%)	Settling Time (s)	Controller and Tuning
Proposed	0.124	8.985	0.248	PID-like FLC and FL
[26]	0.155	4.84	0.2526	FL and GA
[24]	0.21	15	0.64	PID-type FLC and PSO algorithm
[39]	0.40	1.21	0.61	PID and GA
[40]	0.1790	1	0.2585	Neural network
[25]	0.418	17.4	3.17	PID and PSO algorithm

**Table 5 sensors-20-05323-t005:** Controller motion comparison.

Work	Programming	Motion Profile	Tuning	Controller	Sample Time (s)
Proposed	C/C++	S-curve	Fuzzy Relations	PID-type FLC	0.005
[23]	C/C++	Trapezoidal	Empiric	FLC	0.001
[42]	Matlab/Simulink	S-curve	Empiric	PID	-
[43]	LabVIEW NI	-	Empiric	PID	0.0001
[44]	Matlab/Simulink	-	Pole Assignment	PID	0.060
[4]	-	Trapezoidal	Empiric	P-PI	0.001

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
