# Peer review of "A PID-Type Fuzzy Logic Controller-Based Approach for Motion Control Applications"

_sensors, 2020, doi:10.3390/s20185323_

Round 1

Reviewer 1 Report

Dear authors:

       I'm glad to read this paper and giving the comments with my pleasure.

       In this paper, you've introduced the “PID type fuzzy controller strategy” and combined it with the S-curve velocity profile. The controller is carried out online. The algorithm is programming by C code. This is a good control algorithm and implementation. You give some detailed explanations of your concepts in the content. The design flow chart(figure 6)of the controller is  good. The results and comparison description are also explicit.

       Here I give you some suggestions:

       1. In the "Abstract" part, the experiment results details can be expressed in the "part 5".

       2. In the introduction, some references should not be listed if they are not related to your main content or your control algorithm in the paper.

       3. Your English language is good but some minor spell checks also required  such as in "line 90".

       4. Your block diagrams are good in the paper but here I have a question why the "integral constant Ki" can not be used the "FLC tuning strategy"?  you should give the explanaion.

       5. You can change the order or sequence of the Figures and tables to match your content. For example you can arrange figure 3 and table 1 in one page corresponding to your content. Or they will be in different pages and make some mess.

Author Response

REVIEWER 1

Dear authors:

       I'm glad to read this paper and give the comments with my pleasure.

       In this paper, you've introduced the “PID type fuzzy controller strategy” and combined it with the S-curve velocity profile. The controller is carried out online. The algorithm is programming by C code. This is a good control algorithm and implementation. You give some detailed explanations of your concepts in the content. The design flow chart(figure 6)of the controller is good. The results and comparison description are also explicit.

       Here I give you some suggestions:

  1. aca" part, the experiment results details can be expressed in the "part 5".

Thank you for the advice, we added:

These results presented in part 6 demonstrate the performance of the controller since the rise time and settling time are improved according to the state of the art.

  1.     In the introduction, some references should not be listed if they are not related to your main content or your control algorithm in the paper.

 We appreciate your comment, some references related to the control architecture proposed were added

  1. Your English language is good but some minor spell checks also required  such as in "line 90".

Thank you for the observation, the idea was not clear. We rewrite the sentence to:

A fuzzy set is a generalization of a classical set which allows elements of a universal set to belong with a certain degree to a subset.

  1. Your block diagrams are good in the paper but here I have a question why the "integral constant Ki" can not be used the "FLC tuning strategy"?  you should give the explanation.

We appreciate you response, the next sentence is added to satisfy the above question:

 Variables Kp and Kd offer a fast response to the plant, the term Ki is constant since it is only required to reduce the error to zero when the steady state is reached.

  1. You can change the order or sequence of the Figures and tables to match your content. For example you can arrange figure 3 and table 1 in one page corresponding to your content. Or they will be in different pages and make some mess.

We appreciate your comment, we addressed this comment on the paper.

Reviewer 2 Report

The article proposes a PID-FLC for motion control applications. The proposed controller is applied to a linear platform in order to test its performance. A scheme similar to the proposed PID type FLC can be found in several works. See the literature and update the list of references accordingly.

Put the minus signal “-” in the feedback control system of Figure 2. It is a negative feedback system.

Replace the word “compounded” used throughout the text by a more adequate one like “composed”, or similar. In fact, the manuscript needs to be revised regarding the writing.

Complete the decision block of Figure 6 with the labels “Yes” and “No”.

Cite Figures 7, 9, 14 in the text.

In section 5, give the value of integral gain Ki used in experiments.

Cite Figure 12a) in the text. Correct legend of Figure 12b) to “control signal” instead of “error”.

Cite Figure 15a) in the text. Correct legend of figure 15b) to “control signal” instead of “error”.

The authors emphasize the fact of using the C/C++ language for implementing the PID fuzzy logic system, but no details are given about the development of the PID-FLC on C/C++. Provide these details referring how the performance is improved by using this kind of language.

Author Response

REVIEWER 2

The article proposes a PID-FLC for motion control applications. The proposed controller is applied to a linear platform in order to test its performance. A scheme similar to the proposed PID type FLC can be found in several works. See the literature and update the list of references accordingly.

Some similar papers were cited in the manuscript to carry out with your suggestion, Thank you.

  1. Put the minus signal “-” in the feedback control system of Figure 2. It is a negative feedback system.

Thank you for your correction, Figure 3 was modified.

  1. Replace the word “compounded” used throughout the text by a more adequate one like “composed”, or similar. In fact, the manuscript needs to be revised regarding the writing.

We appreciate your suggestion, we rewrite some sentences in the paper.

  1. Complete the decision block of Figure 6 with the labels “Yes” and “No”.

We appreciate the observation, the flow chart of Figure 7 was modified

  1. Cite Figures 7, 9, 14 in the text.

We appreciate this observation; Figures were cited in the manuscript.

  1. In section 5, give the value of integral gain Ki used in experiments.

Thank you for your suggestion, the Ki values are added on the paper

  1. Cite Figure 12a) in the text. Correct legend of Figure 12b) to “control signal” instead of “error”.

The legend has been updated

  1. Cite Figure 15a) in the text. Correct legend of figure 15b) to “control signal” instead of “error”.

The legend has been updated

  1. The authors emphasize the fact of using the C/C++ language for implementing the PID fuzzy logic system, but no details are given about the development of the PID-FLC on C/C++. Provide these details referring how the performance is improved by using this kind of language.

We appreciate your comment, we added the next paragraph on text:

The idea of using a free programming language, that does not need licenses, for the design of motion control systems makes the system affordable and easy to apply in other programming languages and even different embedded systems. In this paper, three C/C++ functions were created to compute the motion profile, the membership functions and PID-type FLC respectively. The real structure of the motion control algorithm is depicted in Table 5.

Reviewer 3 Report

The paper is of interesting and up-to-date topic. Although the problem of linear motion control has been discussed for several years the problem is still important and new more effective solutions for this purpose are still developed and presented in current research papers.

In this manuscript the authors propose a new control algorithm that is mainly based on the fuzzy logic and that can be used for motor motion control. The provided experiments prove the efficiency of the proposal. Therefore, when taking into account the scientific value – the paper is worth of publishing.

In overall – the paper is of good structure and proper length and was prepared with necessary care. The proposed controller has been described clearly and supported by necessary and exact results of provided experiments. Therefore I recommend to publish this paper. In fact, I have only some minor remarks that can help to improve your paper:

  • I suggest to change a bit the title of the manuscript. In my opinion the title “A PID-type Fuzzy Logic Controller for Linear Motion Control Applications” would be more suitable.
  • Although the literature review has been done properly I suggest to extend it a bit. In particular I suggest to show the popularity and usefulness of fuzzy logic not only in motion control problems but also in many industrial and non-industrial purposes such as: energy distribution and management (e.g.: 10.1016/j.jclepro.2020.123810; 10.1007/978-981-15-6259-4_62), demand forecasting (e.g.: 10.1007/978-3-030-51156-2_116), information resource management (e.g.: 10.1007/978-3-030-54215-3_11), predictive maintenance (e.g.: 10.1007/978-3-030-53651-0_21), material handling (e.g.: 10.1016/j.neucom.2018.05.125) or even supply chain management (see. eg.: 1016/j.ijpe.2020.107883).
  • I suggest to carefully read the whole manuscript before publication. Although it was prepared with necessary care – there are some language and editorial errors that must be removed (see. e.g. – line 230 – “it can be observe…”).

Author Response

REVIEWER 3

The paper is of interesting and up-to-date topic. Although the problem of linear motion control has been discussed for several years the problem is still important and new more effective solutions for this purpose are still developed and presented in current research papers.

In this manuscript the authors propose a new control algorithm that is mainly based on the fuzzy logic and that can be used for motor motion control. The provided experiments prove the efficiency of the proposal. Therefore, when taking into account the scientific value – the paper is worth of publishing.

In overall – the paper is of good structure and proper length and was prepared with necessary care. The proposed controller has been described clearly and supported by necessary and exact results of provided experiments. Therefore I recommend to publish this paper. In fact, I have only some minor remarks that can help to improve your paper:

  • I suggest to change a bit the title of the manuscript. In my opinion the title “A PID-type Fuzzy Logic Controller for Linear Motion Control Applications” would be more suitable.

We appreciate your recommendation, we decided to use your proposedA PID-type Fuzzy Logic Controller-based Approach for Motion Control Applications

  • Although the literature review has been done properly I suggest to extend it a bit. In particular I suggest to show the popularity and usefulness of fuzzy logic not only in motion control problems but also in many industrial and non-industrial purposes such as: energy distribution and management (e.g.: 10.1016/j.jclepro.2020.123810; 10.1007/978-981-15-6259-4_62), demand forecasting (e.g.: 10.1007/978-3-030-51156-2_116), information resource management (e.g.: 10.1007/978-3-030-54215-3_11), predictive maintenance (e.g.: 10.1007/978-3-030-53651-0_21), material handling (e.g.: 10.1016/j.neucom.2018.05.125) or even supply chain management (see. eg.: 1016/j.ijpe.2020.107883).

Thank you for your advice, we added some of the recommended works provided by you:

There are a vast number of applications of FL, such as simplified control of robots [10], control of car engines [11], cruise-control for automobiles [12], prediction systems for early recognition of earthquakes [13], anti-lock braking systems [14], renewable energy systems [15], aircraft engines [16], energy allocation [17,18], demand forecasting [19], predictive maintenance [20], material handling [21], just for mention ones.

  • I suggest to carefully read the whole manuscript before publication. Although it was prepared with necessary care – there are some language and editorial errors that must be removed (see. e.g. – line 230 – “it can be observe…”).

We appreciate your observation, we rewrite some sentences we identify with some grammatical errors.

 Ex: In Figure 9(a) it can be observed that the controller presents a fast response since the system reaches the steady-state in 0.248s and the overshoot is 8.985%. The error signal is shown in Figure 9(b) and it tends to zero.

Round 2

Reviewer 2 Report

The authors have attended to all major issues proposed by the reviewers.